

# Prevalence and incidence of sarcopenia and low physical activity among community-dwelling older Thai people: a preliminary prospective cohort study 2-year follow-up

Kornanong Yuenyongchaiwat[1,2] and Chareeporn Akekawatchai[3,4]

[1] Department of Physiotherapy, Faculty of Allied Health Sciences, Thammasat University, Klong Luang, Pathumthani, Thailand
[2] Thammasat University Research Unit in Physical Therapy in Respiratory and Cardiovascular Systems, Thammasat University, Klong Luang, Pathumthani, Thailand
[3] Department of Medical Technology, Faculty of Allied Health Sciences, Thammasat University, Klong Luang, Pathumtani, Thailand
[4] Thammasat University Research Unit in Diagnostic Molecular Biology of Chronic Diseases related to Cancer (DMB-CDC), Klong Luang, Pathumtani, Thailand

Corresponding author
Kornanong Yuenyongchaiwat,
ykornano@tu.ac.th

## ABSTRACT

**Background**. Sarcopenia, defined as a loss of muscle mass, has become a major health problem in older people. Few prospective studies report the incidence and risk of sarcopenia. Therefore, this study aimed to explore the prevalence of sarcopenia at the baseline and follow-up after 2 years in community-dwelling older Thai individuals.

**Methods**. In 2019, 330 older people were recruited from a community-dwelling population, and these participants were requested to present again in 2021. Sarcopenia was diagnosed using the criteria for the Asia Working Group for Sarcopenia (AWGS). All participants were asked to perform a 6-meter walk test, handgrip strength test, and bioelectric impedance assessment, and complete the Global Physical Activity Questionnaire.

**Results**. The study found that the prevalence of sarcopenia was 65 (19.70%) in 330 older people in 2019, and 44 of 205 participants (21.46%) were reported to have sarcopenia after 2 years. The incidence of sarcopenia was noted to be 2.44% in 2021. Analysis with ANOVA and pairwise comparisons showed that the reversibility of sarcopenia was attributed to high level of physical activity in the 2-year follow-up group ($p = 0.014$, 95% CI [$-1753.25$–$-195.49$]). Further, participants with moderate and high physical activity had a reduced incidence of sarcopenia (odds ratio = 9.00 and 14.47, respectively). Therefore, low physical activity in older people led to the development of sarcopenia from the baseline to the 2-year follow-up, indicating that increased physical activity may be useful in reversing sarcopenia, as suggested in the 2-year follow-up study. Low physical activity could be a risk factor for the incidence of sarcopenia. Hence, the prevention of sarcopenia could promote health improvement through moderate to high physical activity.

## INTRODUCTION

According to the World Population Ageing 2020 Highlights, there are 727 million people aged 65 years or over in the world, with women living longer than men (*United Nations Department of Economic and Social Affairs, Population Division, 2020*). Moreover, this is expected to increase to over 1.5 billion by 2050 (*United Nations Department of Economic and Social Affairs, Population Division, 2020*), and an increasing in the aged population will be noted in all regions, including Thailand. Decline in physical performance can be found in people of advanced age, which limits the ability to perform daily activities and leads to a poor quality of life. Recently, several studies have focused on sarcopenia among older people in European as well as Asian countries. Sarcopenia has been defined as the loss of muscle mass, low muscle strength, and poor physical performance (*Chen et al., 2014a*; *Chen et al., 2020b*; *Cruz-Jentoft et al., 2019*; *Da Silva Alexandre et al., 2014*). The prevalence of sarcopenia has been reported to be different among older adults depending on the region (*i.e.,* Asian or European countries), diagnostic criteria (*e.g.,* European Working Group on Sarcopenia in Older People (EWGSOP)), living community (*i.e.,* community-dwelling, nursing home, or hospitalized participants), or the method of measurement of muscle mass (*e.g.,* the Bio-electrical Impedance Analysis (BIA), the dual-energy X-ray absorptiometry (DEXA), and anthropometrics). This leads to differences in the reported incidence of sarcopenia in older people (*Lee et al., 2013*; *Papadopoulou et al., 2020*; *Shafiee et al., 2017*). Recently, a systematic review with a meta-analysis of 30,287 individuals in the community revealed that the prevalence of sarcopenia in adults aged 60 years or over was 11% in men and 9% in women (*Papadopoulou et al., 2020*). However, prevalence rates using the AWGS definition vary from 6.4% to 41.3% in the older community-dwelling population (*Chang et al., 2020*; *Han et al., 2016*). For example, the prevalence of sarcopenia among Korean adults aged ≥ 65 years was 40.3% in men and 41.3% in women according to the AWGS definition (*Chang et al., 2020*), whereas the prevalence of sarcopenia in Chinese suburb-dwelling adults aged ≥ 60 years was 6.4% in men and 11.5% in women (*Han et al., 2016*). Several studies have reported the prevalence of sarcopenia in a cross-sectional study design; however, few studies have been conducted with a prospective design to report the data of follow-up sessions. The present study tested the hypothesis that the incidence of sarcopenia has increased in the 2-year follow-up. Therefore, the present study aimed to explore the prevalence of sarcopenia in 2019 with a follow-up after 2 years.

## MATERIALS & METHODS

All individuals given their written informed consent, and the institute's committee on human research approved the study protocol. This study protocol was reviewed and approved by the Ethics Human Committee of Thammasat University based on the Declaration of Helsinki, the Belmont Report, CIOMS Guidelines, and the International Practice (ICH-GCP), approval number COA no. 023/2562. The clinical trial registration is TCTR20190218002.

A prospective cohort study was designed with older people. Based on a previous study conducted in Thailand, the prevalence of sarcopenia was 30.5% in community-dwelling

older Thai older adults (*Khongsir et al., 2016*). Therefore, sample size calculations indicated that 330 older adults were needed to complete the study, with males and females aged $\geq$ 60 years. Older people who lived in community-dwelling setting were invited to participate, screened for sarcopenia during community health service in 2019, and reassessed in 2021.

Sarcopenia was defined as per the criteria of the AWGS 2019, which is composed of gait speed, handgrip, and low muscle mass (*Chen et al., 2020b*). To evaluate gait speed, a 6-meter walk test was performed. The participants were instructed to walk for 6 m at a comfortable speed over a flat and straight surface three times, and the average velocity was calculated as distance divided by time. Participants who had a gait speed of <1.0 m/s were categorized as having slow gait speed. Handgrip strength measured the physical performance; all participants were asked to perform the test three times using a handgrip dynamometer (model TKK, JAPAN) in the dominant hand, and the maximal grip strength value was recorded. Poor grip strength was defined as <18 kg in women and <28 kg in men. Low muscle mass was measured using BIA. (Omron HBF-375 body composition monitor; Omron Healthcare Co., Ltd., Japan). BIA has been accepted for screening sarcopenia under the guidelines of the AWGS (*Chen et al., 2014a*; *Chen et al., 2020b*). In addition, several studies report that BIA is a favorable alternative to magnetic resonance imaging and DEXA as a screening tool for sarcopenia or skeletal muscle mass (*Aleixo et al., 2020*; *Fujimoto et al., 2018*; *Janssen et al., 2000*; *Lee et al., 2018*). The percentage of skeletal muscle mass (%SMM) was displayed in the BIA, and the skeletal muscle mass (SMM) was calculated as %SMM multiplied by body weight (kg). Finally, skeletal muscle mass index (SMI) was analyzed by multiplying SMM by height in meters$^2$. Low SMI was determined as <7.0 kg/m$^2$ in male and <5.7 kg/m$^2$ in female participants (*Chen et al., 2020b*). According to the AWGS 2019, sarcopenia is categorized as no sarcopenia, a possible sarcopenia (defined by low muscle mass or poor physical performance), sarcopenia (defined by low physical performance or slow gait speed plus low SMI), and severe sarcopenia (defined by low physical performance, slow gait speed, and low SMI).

The Global Physical Activity Questionnaire (GPAQ), developed by the World Health Organization, was administered to all individuals to measure physical activity levels. The three domains of the GPAQ are activity at work, traveling to and from places, and recreational activities (*Armstrong & Bull, 2006*). The GPAQ is divided into three categories: low level of physical activity (<600 metabolic equivalent of task (MET) minutes per week), moderate (600–1499 MET*min*wk$^{-1}$), and high ($\geq$1,500 MET*min*wk$^{-1}$) (*Armstrong & Bull, 2006*).

Again, all participants were invited for a follow-up 2 years later (*i.e.,* in 2021); however, only 205 participants who were able to walk or move agreed to the re-assessment of physical performance, gait speed, and muscle mass. A total of 125 individuals did not participate in person; therefore, personal contact or telephone was used to follow their health status (*e.g.*, data on history of illness and physical limitation).

Statistical analyses were performed using IBM SPSS version 23. Descriptive statistics with mean, standard deviation, and frequency were presented. The $t$-test or chi-square test was used to compare participants with and without sarcopenia based on the AWGS 2019 guidelines at the initial assessment and after 2 years. ANOVA was used to compare

risk factors for sarcopenia over a 2-year period (*i.e.,* no change in sarcopenia, new cases, and reversibility of sarcopenia). Finally, logistic regression analysis was used to determine independent predictors for developing the reversibility of sarcopenia. Statistical significance was set at a *p*-value of <0.05.

## RESULTS

A total of 330 older people were recruited in 2019; however, only 205 individuals (62.12%) attended the follow-up in 2021 (25.25 month). In the initial assessment, of the 330 older people in 2019, 65 (19.70%) had sarcopenia and 265 (80.30%) had no sarcopenia. The prevalence of sarcopenia over a 25-month follow-up was 44 (21.46%); where 37 (18.05%) individuals had sarcopenia and seven (3.41%) individuals had severe sarcopenia. In addition, 161 (78.54%) individuals were defined as having no sarcopenia; 16 (7.80%) individuals had a possibility of developing sarcopenia (see Fig. 1). Tables 1 and 2 display the characteristic data of older Thai people at baseline and the follow-up sessions. Furthermore, the McNemar test was analyzed and it was found that older people had significant change in sarcopenia at two time points ($p = 0.002$). In the participants who agreed for re-assessment, significant decrease in muscle strength ($22.22 \pm 6.29$ kg. *vs.* $21.52 \pm 5.12$ kg; $p = 0.044$) and low muscle mass ($6.03 \pm 1.38$ kg/m$^2$ vs. $5.89 \pm 1.11$ kg/m$^2$; $p = 0.023$) were observed in the follow-up study; but no change in physical performance ($1.18 \pm 0.22$ m/s *vs.* $1.21 \pm 0.59$ m/s; $p = 0.547$) was observed.

Regarding the reasons for failure to follow-up, of the 125 participants who did not participate in the follow-up study (37.87% of 330 participants in the initial study in 2019), 50 (14.29%) did not respond or contact; of these, 43 participants were in the no sarcopenia group. In addition, 27 (8.18%) individuals were weak or had difficulties moving or walking outside the house, and one (0.30%) who had been categorized to have sarcopenia had died (see Fig. 2). Interestingly, chi-square test of significance indicated that there was a significant difference in the proportion of individuals affected by sarcopenia in the initial study and the reason for participating and non-participating after the 2-year follow-up ($\chi^2$ =26.347, $p < 0.001$).

In addition, participants with sarcopenia had lower physical activity than those without sarcopenia, both in the initial assessment and the follow-up. Interestingly, more new cases of sarcopenia were observed in individuals with low physical activity ($472.00 \pm 754.40$ MET*min*wk$^{-1}$) than in those with moderate ($1263.41 \pm 1552.06$ MET*min*wk$^{-1}$) to high physical activity ($2237.78 \pm 2142.02$ MET*min*wk$^{-1}$) (see Table 3). To determine the importance of physical activity for reversibility of sarcopenia, individuals with moderate physical activity had an odds ratio (OR) of 9.00 (95% confidence interval (CI) 1.05–77.04, $p = 0.045$) and high physical activity had an OR of 14.47 (95%CI [1.82–115.38], $p = 0.012$) for developing reversibility of sarcopenia compared to those with low physical activity (see Table 4). Moderate to high physical activity levels were related to a reduced incidence of sarcopenia events.

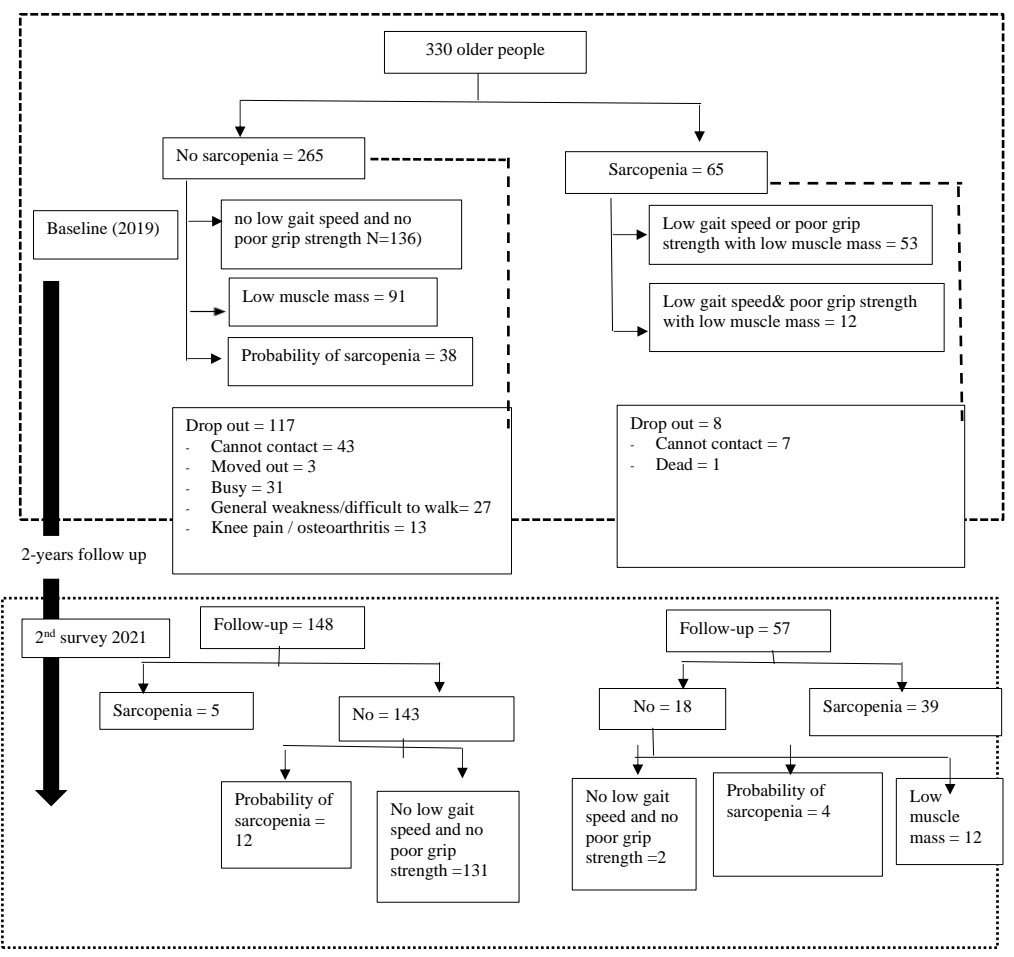

**Figure 1** Sarcopenia category by diagnostic algorithms of the Asian Working Group for Sarcopenia.

**Table 1 Characteristics of Thai older adults who completed an initial study among non-sarcopenia and sarcopenia based upon AWGS 2019 (N = 330).**

| | Total (N = 330) | Sarcopenia (n = 65) | Non-sarcopenia (n = 265) | 95% CI | p-value |
|---|---|---|---|---|---|
| Age (yrs) | 66.85 ± 5.53 | 70.23 ± 6.39 | 66.03 ± 4.99 | 2.77 to 5.64 | <0.001 |
| **Sex** [*] | | | | | 0.170 |
| Female | 265 | 46 | 205 | | |
| Male | 65 | 19 | 46 | | |
| Skeletal muscle index (kg/m$^2$) | 6.11 ± 1.30 | 5.05 ± 0.87 | 6.37 ± 1.25 | −1.64 to −1.00 | <0.001 |
| Gait speed (m/s) | 1.20 ± 0.23 | 1.10 ± 0.24 | 1.22 ± 2.22 | −0.18 to −0.06 | <0.001 |
| Handgrip strength (kg) | 22.69 ± 6.18 | 18.55 ± 3.83 | 23.70 ± 6.22 | −6.74 to −3.56 | <0.001 |
| Physical activity (MET*min*wk$^{-1}$) | 2698.21 ± 3720.35 | 1570.77 ± 3181.80 | 2974.75 ± 3795.37 | −2407.01 to −400.96 | 0.003 |

Notes.
*Analysis by chi-square test.

**Table 2** Characteristics of Thai older adults who completed baseline and follow-up sessions among non-sarcopenia and sarcopenia over a 2 year follow-up ($N = 205$).

| | Total ($N = 205$) | Sarcopenia ($n = 44$) | Non-sarcopenia ($n = 161$) | 95% CI | *p*-value |
|---|---|---|---|---|---|
| Age (yrs) | $70.02 \pm 5.86$ | $72.30 \pm 6.48$ | $68.63 \pm 5.24$ | 1.82 to 5.52 | <0.001 |
| **Sex**[*] | | | | 3.197 | 0.109 |
| Female (156) | 156 (76.10%) | 29 (18.59%) | 127 (81.41%) | | |
| Male (49) | 49 (23.90%) | 15 (30.61%) | 34 (69.39%) | | |
| Skeletal muscle index (kg/m$^2$) | $5.89 \pm 1.11$ | $5.17 \pm 0.69$ | $6.01 \pm 1.11$ | $-1.27$ to $-0.57$ | <0.001 |
| Gait speed (m/s) | $1.21 \pm 0.59$ | $0.99 \pm 0.20$ | $1.27 \pm 0.65$ | $-0.48$ to $-0.09$ | 0.005 |
| Handgrip strength (kg) | $21.52 \pm 5.12$ | $19.78 \pm 4.11$ | $22.00 \pm 5.27$ | $-3.90$ to $-0.52$ | 0.011 |
| Physical activity (MET*min*wk$^{-1}$) | $1324.93 \pm 1613.35$ | $785.77 \pm 1143.24$ | $1478.30 \pm 1700.79$ | $-1228.86$ to $-156.20$ | 0.021 |

**Notes.**
[*]Analysis by chi-square test.

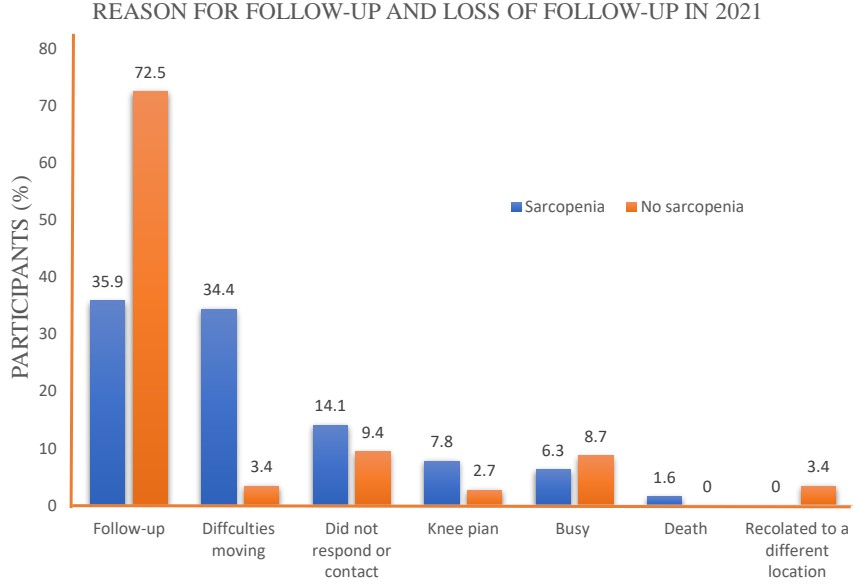

**Figure 2** Reasons for participating the study in 2021.

## DISCUSSION

The present study explored the prevalence of sarcopenia according to the AWGS 2019 definition in 2019, and then re-assessed it in 2021 in a 2-year follow-up study, observing that the prevalence of sarcopenia was 19.70% among 330 older adults in 2019 and 21.46% in 2021. Additionally, the incidence of sarcopenia over 2 years was 2.44%. This means that the incidence of sarcopenia increased by 1.20% every year. Interestingly, older participants with sarcopenia who did not attend the follow-up after a 2-year period showed a decrease in physical performance or adverse health outcomes in 2021, and conversely, participants who had a low physical performance or adverse health outcomes in 2019 had failed or did not attend the follow-up re-assessment after 2 years.

**Table 3  Compare of risks factors for the prevalence, incidence and reversibility of sarcopenia.**

| | Sarcopenia after 2 years period | | | 95% CI, (*p*-value)[a] | 95% CI, (*p*-value)[b] | 95% CI, (*p*-value)[c] |
|---|---|---|---|---|---|---|
| | Same (*n* = 182) | Reversibility (*n* = 18) | New cases (*n* = 5) | | | |
| PA (MET*min*wk$^{-1}$) | 1263.41 ± 1552.06 | 2237.78 ± 2142.02 | 472.00 ± 754.40 | −1753.25 to −195.49 (*p* = 0.014) | −637.58 to 2220.39 (*p* = 0.276) | 172.21 to 3359.34 (*p* = 0.030) |

**Notes.**
[a] Compare between no change in sarcopenia and reversibility of sarcopenia.
[b] Compare between no change in sarcopenia and new case of sarcopenia.
[c] Compare between reversibility of sarcopenia and new case of sarcopenia.

**Table 4  Results of logistic regression analysis for physical activity for reversibility of sarcopenia over 2-year follow-up.**

| | Odds ratio | 95% CI for OR | *p*-value |
|---|---|---|---|
| GPAQ | Reference low physical activity | | |
| Moderate physical activity | 9.00 | 1.051–77.035 | 0.045 |
| High physical activity | 14.47 | 1.816–115.378 | 0.012 |

**Notes.**
CI, confidence interval; OR, odds ratio; GPAQ, Global Physical Activity Questionnaire.

In Thailand, the prevalence of sarcopenia was 10.0%, based on the AWGS 2019 and AWGS 2014 criteria (*Therakomen, Petchlorlian & Lakananurak, 2020*). In our study, the prevalence of sarcopenia was 16.1% according to the AWGS 2014 criteria (*Yuenyongchaiwat & Boonsinsukh, 2020*) and 19.7% according to the AWGS 2019 criteria. Similarly, the prevalence of sarcopenia among community-dwelling older people in West China was reported to be 19.3% and 22.8% according to the AWGS 2014 and AWGS 2019 criteria, respectively (*Liu et al., 2021*). According to the AWGS 2019 criteria, poor grip strength in male participants (defined as <28 kg) and gait speed in both male and female participants (defined as <1.0 m/s) were increased compared with the criteria in 2014 (26 kg for grip strength for male and <0.8 m/s for both male and female); therefore, that is the reason for the increase in the prevalence of sarcopenia: using the AWGS 2019 definition compared to the AWGS 2014 definition.

The incidence of sarcopenia was 2.44% in older Thai people and the average incidence of sarcopenia over 2 years was calculated to be 1.20% annually. Similarly, the incidence of sarcopenia from baseline to 2 years was 6.9% in Chinese men and women aged 65 years and older, and the average annual incidence of sarcopenia over 4 years was 3.1% (*Yu et al., 2014*). Comparably, the study also found that reduced physical activity was associated with the occurrence of sarcopenia at the baseline and 2-year follow-up. *Yu et al. (2014)* reported that the risk factors for incidence of sarcopenia were low levels of physical activity and body mass index. Therefore, increasing physical activity levels (*i.e.,* moderate to high physical activity level) can reduce the incidence of sarcopenia.

As mentioned previously, among the participants who did not participate in 2021, 41(32.8%) individuals did not participate due to declined physical performance (*e.g.,* general weakness, inability to walk outside, knee pain, dead) or slow gait speed (knee pain).

Therefore, it seems that over a quarter of participants in the study (32.8% of 125) who did not come for follow-up had a decline in their health, such as knee pain and inability to walk outside the house. Hence, it is suspected that over the 2-year period, older people had a decline in their health status and an incidence of adverse health events, including sarcopenia. Furthermore, some participants (*e.g.*, lack of physical activity) showed a clinical change from non-sarcopenia to possible development of sarcopenia in the future. Thus, future studies should explore and follow their health outcomes.

This is the first study exploring the incidence of sarcopenia among older Thai people; however, some limitations should be noted. The number of participants who participated in the follow-up study was relatively less. This might be due to the COVID-19 pandemic, which might have affected participation. In December 2019, an outbreak of COVID-19 occurred in Thailand. As a consequence, Thailand has been severely impacted and has continued to experience rising infection rates according to reports. The participants' follow-up assessment was during the COVID-19 pandemic. Only healthy older adults who were able to walk or move participated in the follow-up study. Therefore, the prevalence and incidence of sarcopenia cannot be generalized for all Thai people. Another limitation is that there were higher numbers of female than male participants at both the initial assessment and the follow-up. Thus, the results regarding the incidence of sarcopenia should be interpreted with caution.

## CONCLUSIONS

The prevalence of sarcopenia in community-dwelling older Thai people was 19.70% among 330 older adults in 2019 and 21.46% among 205 older people in 2021; indicating that the incidence of sarcopenia was 2.44% in 2-years. In addition, increasing physical activity to a moderate-high level in older people can lead to reduced risk of sarcopenia from baseline to the 2-year follow-up.

## ACKNOWLEDGEMENTS

The authors would like to thank the participants in Pathumtani community and their caregivers for participating in the study.

### Funding
This study was fully supported by the Office of the Higher Education Commission and the Thailand Research Fund (contract no. NRCT5-RSA63010-04). The funders had no role in study design, data collection and analysis, decision to publish, or preparation of the manuscript.

### Grant Disclosures
The following grant information was disclosed by the authors:
The Office of the Higher Education Commission and the Thailand Research Fund: NRCT5-RSA63010-04.
## Competing Interests

The authors declare there are no competing interests.

## Author Contributions

- Kornanong Yuenyongchaiwat conceived and designed the experiments, performed the experiments, analyzed the data, prepared figures and/or tables, authored or reviewed drafts of the paper, and approved the final draft.
- Chareeporn Akekawatchai performed the experiments, analyzed the data, prepared figures and/or tables, authored or reviewed drafts of the paper, and approved the final draft.

## Human Ethics

The following information was supplied relating to ethical approvals (i.e., approving body and any reference numbers):

The Ethics Human Committee of Thammasat University.

## Data Availability

The raw measurements are available in the Supplementary File.

## Supplemental Information

Supplemental information for this article can be found online at http://dx.doi.org/10.7717/peerj.13320#supplemental-information.

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
