# Peer review of "Prevalence and incidence of sarcopenia and low physical activity among community-dwelling older Thai people: a preliminary prospective cohort study 2-year follow-up"

_PeerJ, doi:10.7717/peerj.13320_

## Round 0.1 · original submission · Major Revisions

Dr Yuenyongchaiwat,

Congratulations on a well-conducted study. However, I have received the reviews of your manuscript "Prevalence and incidence of sarcopenia and low physical activity among community-dwelling older Thai people during COVID-19 pandemic: A preliminary prospective cohort study 2-year follow-up " and based upon the reviewer's comments your manuscript requires major revisions.

Please address all of the reviewer's comments in a timely manner.

Thanks, A/Prof Mike Climstein

Reviewer 1 ·

Basic reporting

The research question is well warranted and relevant. It is also well-defined and meaningful to the older adult population. Well done on examining an area that has been underdeveloped and bringing to fruition an area that will make a significant impact. Major revisions are required as I believe there are sections that need to be more clear and concise, given the importance and significance you state with this research topic. This message of clarity does not come across yet, especially being that this is the first study to investigate sarcopenia in Thia older adults residing in the community.

The areas of interest that need editing are the following:

In terms of your title, recommend removing the COVID-19 pandemic as you do not mention COVID-19 anywhere else in your manuscript.

The reference list needs to be screened through as currently does not follow a consistent PeerJ style.

Table formatting needs to be cleaned up (provided comments in the attachment).

Experimental design

Materials and methods
How many participants were invited to participate and screened in 2019, this was not stated in the manuscript
In terms of the walk test, what did participants do to complete this? using what equipment and instructions were provided to the participants - as you include relevant information for handgrip strength and BIA but this is missing for the walk test.

In the results you mention severe sarcopenia please provide in the introduction what this terminology means and the significance of this term.

Validity of the findings

The discussion needs some work on the delivery of the results and the importance of the study being brought up earlier (see attached file). the relevance of stating that this is the first study to explore the incidence of sarcopenia among Thai older adults needs to be driven and consistent throughout the manuscript.

Additional comments

Please see the attached file that has been scanned which provides more comments around spelling, grammar, tense and edits to your manuscript.

Annotated reviews are not available for download in order to protect the identity of reviewers who chose to remain anonymous.

Reviewer 2 ·

Basic reporting

This manuscript is written well, however, this need to be revised a few points.
1) Please describe the hypothesis of this study more specifically.
2) Figures are hard to understand. Also, it is not clear. Please correct.

Experimental design

no comment'

Validity of the findings

no comment'

Additional comments

no comment'

Reviewer 3 ·

Basic reporting

1. The lines in Figure 1 should be revised to be standard.
2. Line 160-162 cannot be supported by Table 3, please explain or revise.
3. The language should be revised throughout the manuscript with help of native English speakers.

Experimental design

1. As shown in line 111-114, Sarcopenia diagnosis and classification should be checked and corrected. Based on AWGS 2019, severe sarcopenia is diagnosed as low handgrip strength + slow gait speed and low SMI. Your definition of possible sarcopenia is also incorrect.
2. As the physical activity is mostly associated with novel findings of this study, more details regarding the assessed results by Global Physical Activity Questionnaire should be reported, such as exercise program, intensity, and frequency of included participants.
3. The authors only reported changed prevalence rates of sarcopenia in two time points, however, the significance of these difference should be assessed with appropriate statistic methods, such as McNemar tests. Besides, the changes in handgrip strength, gait speed and SMI across two time points should also be assessed using paired sample T tests or Wilcoxon tests.

Validity of the findings

no comment

Additional comments

This study was well designed but with a small sample. Although several important findings were reported, such as the increased prevalence of sarcopenia and the roles of physical activity in reversing sarcopenia, these outcomes are in line with expectations. The manuscript may be accepted for publication if the listed problems be elegantly solved.

---

## Round 0.2 · accepted · Accept

Dr Yuenyongchaiwat, and Dr Akekawatchai, thank you for addressing all of the Reviewers' comments in a timely manner. I am pleased to inform you that I am recommending your manuscript for publication to the PeerJ Editor.

Thank you again for submitting your manuscript to PeerJ and we look forward to future submissions from your research. A/Prof Mike Climstein

Reviewer 1 ·

Basic reporting

No further comments. Well done!

Experimental design

No further comments.

Validity of the findings

No further comments.

Additional comments

No further comments.

Reviewer 3 ·

Basic reporting

Well done.

Experimental design

Well done.

Validity of the findings

Well done.

Additional comments

The authors have well revised the manuscript based on comments.